# Multimodal generative AI for automated pavement condition assessment: Benchmarking model performance

Chang Xu[1], Lei Shu[1], Anh Dao[2], Yue Cui[1]*

**1** School of Planning, Design, and Construction, Michigan State University, East Lansing, Michigan, United States of America, **2** Department of Computer Science and Engineering, Michigan State University, East Lansing, Michigan, United States of America

* cuiyue@msu.edu

## Abstract

Accurate and efficient pavement condition assessment is essential for maintaining roadway safety and optimizing maintenance investments. However, conventional assessment methods such as manual visual inspections and specialized sensing equipment are often time-consuming, expensive, and difficult to scale across large networks. Recent advancements in generative artificial intelligence (GAI) have introduced new opportunities for automating visual interpretation tasks using street-level imagery. This study evaluates the performance of seven multimodal large language models (MLLMs) for road surface condition assessment, including three proprietary models (Gemini 2.5 Pro, OpenAI o1, and GPT-4o) and four open-source models (Gemma 3, Llama 3.2, LLaVA v1.6 Mistral, and LLaVA v1.6 Vicuna). The models were tested across four task categories relevant to pavement management: distress and feature identification, spatial pattern recognition, severity evaluation, and maintenance interval estimation. Model performance was assessed across five dimensions: response rate, response correctness, consistency, multimodal errors, and overall computational intensity and cost. Results indicate that MLLMs can interpret street-level imagery and generate task-relevant outputs in a cost-effective manner. Among the evaluated models, we recommend GPT-4o as the preferred option, as it balances responsiveness, accuracy, and computational cost.

## Introduction

Road condition assessment is a key component of transportation infrastructure management. It provides information necessary to evaluate the current state of roadways, schedule maintenance activities, and allocate resources efficiently. Accurate assessments can contribute to improved safety, reduced travel disruption, and lower vehicle operating costs [1,2]. Over time, road condition monitoring also supports infrastructure asset management by informing long-term planning, helping prioritize

**Data availability statement:** We will share data on public GitHub as soon as this paper is accepted.

**Funding:** The project is funded by the Great Lakes–Northern Forest (GLNF) Cooperative Ecosystem Studies Unit (CESU), award number W912HZ-19-SOI-0002. The funder had no role in the study design, data collection and analysis, decision to publish, or preparation of the manuscript. The content of this publication does not necessarily reflect the views or policies of the funder.

**Competing interests:** No authors have competing interests.

maintenance needs, and reducing overall life-cycle costs [3–5]. Beyond structural and safety benefits, effective pavement management also reduces fuel consumption, tire wear, and mechanical damage caused by surface irregularities, offering both economic and environmental advantages [6].

Road condition assessment methods range from structural evaluations that measure subsurface integrity using techniques such as falling weight deflectometer testing [7,8] and ground-penetrating radar [9,10], to surface-level inspections, which analyze visible pavement conditions such as cracking, rutting, raveling, and potholes [11–13]. This study focuses on road surface assessment, which evaluates road condition based on the physical characteristics of the pavement surface.

Conventional road surface assessments have traditionally relied on manual inspection techniques, which are often time-consuming, labor-intensive, and subject to observe variability. In response, recent developments in computer vision and artificial intelligence have enabled the adoption of automated, image-based methods [14–16]. Deep learning [14,17] and other machine learning techniques [18,19] are widely applied in pavement condition assessment to enhance accuracy, consistency, and operational efficiency [18,20,21]. Developing these models requires large volumes of high-quality, accurately labeled training data [22,23]. Acquiring such data is often challenging and time-consuming due to the need for extensive manual annotation [24]. In addition, most models are designed for a single, task-specific function [25]. Since road surface assessment involves multiple stages, including defect detection, severity evaluation, and maintenance prioritization, separate models are typically required to address each component of the workflow.

More recently, advances in generative artificial intelligence (AI), particularly multimodal models, have gained attention for their potential to move beyond traditional support functions and autonomously generate task-relevant outputs with minimal contextual input. Unlike conventional machine learning models that require extensive training datasets and pre-defined rules, generative artificial intelligence (GAI) models can respond adaptively to a range of inputs and generate task-relevant outputs with limited contextual information [26]. Leveraging these capabilities, recent studies have begun to apply MLLMs in urban research, particularly using street-view imagery to analyze built environments, urban form, and neighborhood conditions [27–29].

However, the application of MLLMs using street-view remain for surface-level road condition assessment remains largely unexplored. Pavement evaluation typically involves detecting surface distress, classifying severity, and recommending maintenance actions, which have traditionally required multiple specialized models. With their multimodal understanding and generative capabilities, MLLMs have the potential to integrate these tasks within a single framework and streamline the assessment process. Before MLLMs can be reliably adopted for pavement evaluation using street-view images, it is necessary to benchmark their current capabilities, assess their performance across key road condition assessment tasks, and determine the degree of fine-tuning and adaptation required for reliable application in pavement condition assessment. Beyond establishing baseline benchmarks, this study expands prior work by comparing the performance, reliability, and computational efficiency of

multiple MLLMs. Although earlier studies have explored the use of individual MLLMs with street-view imagery, they primarily focused on demonstrating the potential of a single model. To address this gap, the present study evaluates the capability of multiple MLLMs for surface-level road condition assessment and examines performance variations across key tasks. Both proprietary and open-source models are included to capture a comprehensive view of current technological capacity, as they differ in architecture, training data, and accessibility. Evaluating both types is important because it enables an assessment of whether open-source alternatives can achieve performance comparable to commercial systems, which has implications for cost-effectiveness and the feasibility of large-scale adoption by public agencies and researchers. The analysis also accounts for computational cost, providing insight into the trade-offs between performance and efficiency.

Building on this framework, this study examines the potential of multimodal large language models (MLLMs) for assessing road surface conditions based on the physical state of pavement. The evaluation focuses on four key tasks relevant to pavement management workflows: (1) surface distress and feature identification (e.g., cracks, potholes, and patches), (2) spatial pattern recognition (e.g., whether defects are isolated, uniformly distributed, or concentrated along edges or drainage areas), (3) severity evaluation (estimating the extent or seriousness of observed defects), and (4) maintenance interval estimation (inferring the timing or urgency of potential maintenance needs). These tasks were selected for their direct relevance to operational decision-making within pavement condition assessment frameworks.

Surface distress identification facilitates the initial detection and classification of visible pavement defects [30]. Spatial pattern recognition supports the identification of localized deterioration, helping to inform targeted maintenance interventions and resource allocation [31]. Severity evaluation translates observed pavement defects into standardized condition ratings that guide maintenance prioritization and investment decisions [32,33]. Finally, maintenance interval estimation involves predicting the timing of future repairs based on the current condition of the pavement, supporting long-term maintenance planning [31].

Model performance is assessed across five performance dimensions: response rate, response correctness, consistency, multimodal errors, and overall computational intensity and cost.

## Literature review

Traditionally, road surface condition assessments have relied on manual inspections, structured surveys, and standardized scoring systems based on the physical condition of the pavement [34,35]. While these methods have served as the foundation for pavement and road management, they are labor-intensive, time-consuming, and susceptible to human error [36]. In response to these limitations, scholars and industry practitioners have increasingly advocated for the adoption of automated, sensor-based approaches. These methods improve the objectivity of assessments, reduce reliance on periodic manual inspections, and enable continuous, real-time monitoring of road conditions [35,37,38]. Technologies such as accelerometers, GPS, cameras, and LiDAR are frequently used to capture high-resolution images of surface defects [39–41].

Building on these technological advances, machine learning [18,42] and deep learning techniques [19,43–45] have been widely applied to road surface image analysis. These computational approaches automate condition assessment, reducing the time and cost associated with traditional manual inspections [17,20]. Beyond improving efficiency, they enhance diagnostic accuracy, streamline analytical workflows, and contribute to more effective road infrastructure management [46].

However, these models are not fully autonomous. They typically integrate human expertise into the decision-making process [47,48] and are unable to make independent judgments about condition severity or maintenance needs. Their effectiveness also depends heavily on large, high-quality annotated datasets, which are often computationally intensive to process [49,50] and may be limited or inconsistently available across different contexts [51].

Recent advancements in GAI have opened new avenues for automating assessment processes across various domains [52,53]. Although MLLMs have not yet reached the point of full automation, they demonstrate versatility across

tasks [54]. Traditional machine learning approaches for surface-level road condition assessment are typically designed and trained for a single, well-defined objective such as defect detection [55–57], defect classification [58–60], or surface segmentation [61,62]. In contrast, MLLM can address multiple tasks within a unified framework by combining visual perception with linguistic reasoning [54]. In addition, when baseline performance is insufficient, these models can be fine-tuned to improve task-specific accuracy [63].

MLLMs appear to have technical capabilities that may support road surface condition assessment using street-level images. These models integrate computer vision functions [64] that allow them to detect, classify, and quantify road defects such as cracks, potholes, and other surface irregularities. In addition to visual processing, they are designed to generate descriptive textual outputs from image content [65], which may allow for context-aware evaluations that combine standardized scoring frameworks with natural language reporting. Furthermore, MLLMs may have the potential to interpret road conditions autonomously and produce preliminary maintenance recommendations, thereby reducing the need for expert intervention. In fact, scholars have utilized street-view images to evaluate built environments [29], streetscape quality [66], walkability [28,67], and neighborhood livability [66], demonstrating that MLLMs can interpret complex visual cues associated with urban design and infrastructure. However, these capabilities have not yet been empirically validated within the domain of pavement assessment.

To address this gap, this study evaluates the performance of MLLMs across four tasks relevant to pavement condition assessment. The selection of tasks reflects common practices in surface road condition assessment: (1) surface distress and feature identification [21,34,68,69] (2) spatial pattern recognition (i.e., crack positioning) [70,71] (3) severity evaluation [32,72,73], and (4) maintenance interval estimation [32,74].

Among the selected models, three are proprietary. Gemini 2.5 Pro, developed by Google DeepMind and released in February 2024, incorporates advanced multimodal reasoning capabilities [75]. OpenAI o1 and GPT-4o, both released by OpenAI in May 2024, represent a lightweight and flagship model, respectively. OpenAI o1 prioritizes computational efficiency for vision-language tasks [76], whereas GPT-4o offers enhanced capabilities in visual perception, language understanding, and rapid multimodal generation [77]. The remaining models are open source. Gemma 3, released by Google DeepMind in April 2024, is a publicly accessible and compact model tailored for ease of fine-tuning and adaptation in academic and research settings [78]. Llama 3.2, developed by Meta and released in April 2024, extends the Llama series with improved contextual reasoning and instruction-following [79]. LLaVA v1.6 Mistral, released jointly by UW-Madison and Microsoft in February 2024, integrates the Mistral backbone for visual-language alignment in multimodal reasoning [80]. LLaVA v1.6 Vicuna, also released by UW-Madison, builds on the Vicuna model and focuses on instruction-following in visual question answering [81].

## Methods

### Data

This study used a multimodal dataset comprising street-level images and textual prompts as model inputs, along with benchmark references for performance evaluation.

**Multimodal input.** The input to MLLMs consisted of street road image and textual prompts designed to reflect real-world road condition assessment tasks. The images were sourced from Google Street View and selected based on PCI records, as detailed in the section on benchmark data. To ensure temporal alignment between pavement condition data and Street View imagery, only images captured within one year of a PCI assessment were included. PCI records prior to 2014 were excluded due to the lack of reliably dated images. The dataset size is relatively small primarily because real-world PCI data are scarce and currently available only from the City of San Francisco. The available dataset contains PCI values and geographic coordinates but does not include street-level imagery, requiring the retrieval of corresponding Google Street View images for each road segment. Furthermore, many segments in San Francisco have perfect PCI scores (i.e., PCI = 100), resulting in limited representation of visible pavement defects. To mitigate potential sampling bias,

a random subset of those records was selected to balance the distribution of pavement conditions. This process resulted in a dataset of 261 road segment images.

Each image was paired with a prompt designed to instruct the model to perform a specific infrastructure-related task. A total of 39 unique prompts were developed, as detailed in S1 Table. These prompts request the identification of observable features (e.g., presence, type, and quantity of cracks or potholes), descriptions of spatial patterns (e.g., clustering near intersections or drainage), assessments of condition severity, and judgments on the urgency of potential maintenance actions. Prompts also ask for explanatory reasoning or justifications. The prompts were written in natural language and structured as multi-part instructions to encourage detailed responses.

**Benchmark data.** To evaluate model outputs, this study used three types of benchmark data: PCI, estimated road maintenance intervals, and manually annotated labels for observable surface features and spatial patterns.

PCI records were obtained from the City of San Francisco's open data portal in February 2025 [82]. The dataset includes street-level information such as PCI assessment dates, numerical scores ranging from 0 to 100, severity categories, road functional classifications, and geographic coordinates. These records were also used to guide image selection, as described in Section 3.1.1.

Road maintenance intervals were estimated by reviewing sequential Google Street View images for visible signs of surface repair. Segments showing repairs within one year of the initial image were labeled as short-term maintenance. Repairs that occurred between one and three years later were labeled as long-term. Segments with no visible repairs within three years were classified as having no maintenance. These labels provide a basis for evaluating the models' ability to estimate repair urgency. While the use of Google Street View imagery may introduce some uncertainty in identifying the exact timing or type of repair, this approach remains a practical solution given the absence of datasets that contain both PCI records and detailed maintenance histories. To minimize potential bias, we aligned image selection with PCI assessment years and verified repairs at the same locations to ensure consistency.

In addition, manual annotations were conducted by members of the research team to support the evaluation of the models' identification capabilities. Each image was independently reviewed by annotators, who answered a series of structured questions corresponding to the model prompts. A total of 29 questions were answered for each image. These questions focused on the presence and type of pavement defects, their spatial distribution, and relevant environmental or contextual conditions. Specifically, annotators assessed (1) surface features, such as whether cracks or potholes were present, and if so, the general type of cracking (transverse, longitudinal, or alligator); (2) spatial distribution patterns, indicating whether defects appeared isolated, spread across the pavement surface, or concentrated along joints, edges, or drainage areas; and (3) environmental and contextual conditions, including visible intersections, drainage features, patched or sealed areas, signs of resurfacing, and evidence of poor drainage or standing water that could exacerbate pavement deterioration. In cases where annotators disagreed, the discrepancies were discussed collectively until a consensus was reached. These annotations serve as benchmarks for evaluating the accuracy of model responses to prompts related to identification tasks.

**Assessment.** This study evaluated the performance of seven MLLMs: Gemini 2.5 Pro, OpenAI o1, GPT-4o, Gemma 3, Llama 3.2, LLaVA v1.6 Mistral, and LLaVA v1.6 Vicuna. Fig 1 shows the general workflow of the MLLM evaluation process. These models were tested across five performance dimensions to assess their ability to complete four categories of road condition assessment tasks of varying complexity. Each image was prompted once, as the 39 prompts were designed primarily as close-ended questions with well-defined terms and clear worded instructions (e.g., yes/no, multiple choice, or numerical rating), which limit open-ended variability in model outputs. The output from each model consisted of text-based responses generated in direct answer to the prompt questions, which were then compared with human-annotated responses to evaluate correctness, consistency, and reasoning quality. All models were configured with a temperature of 0.3 and a top-p of 1.0, while other parameters were kept at their default settings to ensure consistency across platforms. All evaluations were conducted on a high-performance computing cluster equipped with NVIDIA V100 GPUs (32 GB memory) and Intel Xeon Gold 6148 CPUs (2.40 GHz).

 

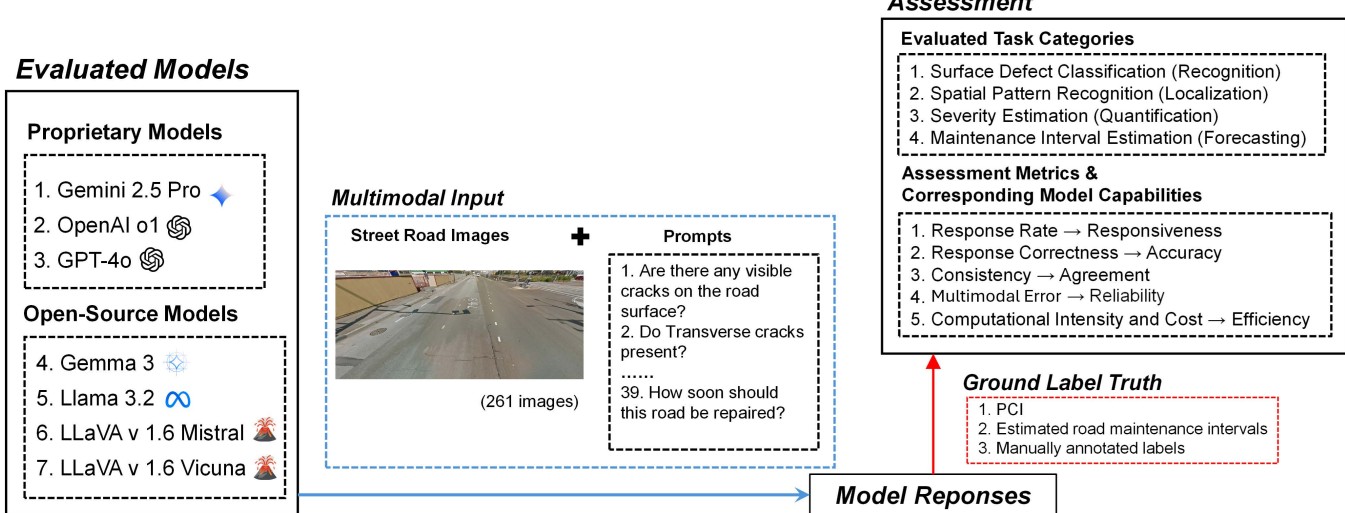

**Fig 1. Workflow of the MLLM evaluation process (Images are used for illustrative purposes only and comply with applicable copyright and licensing requirements; no images are sourced from Google Street View or other proprietary datasets).**

**Task design.** Four task categories, which represent a progression from low-level visual recognition to higher-level condition diagnosis and intervention planning in surface road assessment, were designed to evaluate model capabilities: (1) Pavement surface distress and feature identification, which addresses basic defect detection critical for maintenance prioritization; (2) Spatial pattern recognition, which evaluates whether deterioration is localized or widespread and thus supports infrastructure-level planning; (3) Pavement condition description and severity evaluation, which provides higher-level diagnostic insights into pavement health and deterioration risks; and (4) Maintenance interval estimation, which directly informs scheduling of interventions and long-term resource planning. Table 1 summarizes these four task categories, detailing their associated Task description and assessment focus.

**Assessment dimension.** This study assessed MLLMs' performance across five dimensions, each corresponds to an aspect of model performance: response rate, response correctness, consistency, multimodal errors, and overall computational intensity and cost.

Response rate was used to assess model responsiveness. It was defined as the proportion of prompts for which the model generated a valid and interpretable output. A higher response rate indicates that the model can engage with a wide range of task instructions, while a lower rate suggests difficulty in interpreting or addressing certain prompt types. In this study, the response rate was calculated programmatically as the percentage of images that yielded a valid response for each question. Response rates were first computed for each of the 39 prompts for each model and then aggregated to represent overall responsiveness at the model level.

Response correctness was used to evaluate the factual accuracy of model outputs by comparing them against established benchmark references, including PCI scores, maintenance interval estimation, and manually annotated surface features. Standard classification metrics were calculated, accuracy, precision, recall, and F1 score. These metrics quantify the degree to which model predictions align with ground-truth annotations and serve as objective measures of task performance.

Consistency serves as a measure of inter-model agreement. It is computed using Cohen's Kappa, which evaluates the level of agreement between different models when responding to the same prompt, while adjusting for chance-level concordance. All model responses including uncertain or incomplete ones such as "Not sure" are included in the calculation to reflect real-world deployment conditions.

**Table 1. Task categories and associated prompts.**

| Task Category | Task Detail | Assessment Focus |
|---|---|---|
| Pavement surface distress and feature identification | Identify the presence of cracks; Identify the presence of potholes; Specify the type of crack (e.g., transverse, longitudinal, alligator); Identify surface repair features (e.g., patching, resurfacing); Detect other surface anomalies or treatments. | Detect basic surface defects for maintenance need assessment |
| Spatial pattern recognition | Describe the spatial distribution of cracks (e.g., isolated, widespread, along joints); Describe the spatial distribution of potholes (e.g., near intersections, along edges); Identify whether defects are clustered or evenly distributed; Specify if defects appear near key infrastructure features (e.g., drains, curbs). | Assess spread and clustering to inform localized vs. systemic responses |
| Pavement condition description and severity evaluation | Provide a written description of the overall pavement condition; Assess the severity of observed defects; Estimate the PCI; Describe the visual cues used to justify the severity assessment. | Diagnose pavement health for condition-based decision making |
| Maintenance interval estimation | Recommend a repair timeline (e.g., short-term, long-term, or no maintenance needed). | Translate surface condition into actionable planning guidance |

Multimodal errors were evaluated to identify failures in the model's ability to process, integrate, or reason across visual and textual information, specifically within the context of street-view imagery. This aspect of the analysis focused on the model's reasoning and interpretive behavior, aiming to explore why incorrect or fabricated content was generated. Errors were examined qualitatively to identify recurring patterns and potential underlying causes. As part of this evaluation, hallucinations were included as a specific subtype of multimodal error. In this study, hallucinations were defined as responses that introduced fabricated, unsupported, or irrelevant content not grounded in the input image or prompt. A response was classified as a hallucination when the described feature or condition could not be verified in the corresponding image.

Overall computational intensity and cost are used to evaluate efficiency. These are quantified by recording each model's token usage and response generation time. Statistical significance in model differences is tested using the Kruskal-Wallis test, followed by pairwise Wilcoxon rank-sum tests with Bonferroni correction. To complement these tests, effect sizes were calculated to assess the magnitude and practical significance of the observed differences. For pairwise comparisons, rank-biserial correlation ($r$) was used, and for overall group differences, eta squared ($\eta^2$) was computed based on the Kruskal–Wallis statistic. For proprietary models, token usage is further translated into estimated monetary cost based on publicly available pricing, enabling comparison of cost-efficiency.

## Results

### Response rate

The evaluated MLLMs demonstrated the ability to produce outputs in response to domain-specific prompts related to road surface condition assessment. As shown in Fig 2, the overall average response rate, calculated by aggregating model outputs across four task categories, was 63.65%, indicating a moderate capacity to produce interpretable outputs. Both the highest and lowest response rates were observed among the proprietary models. Gemini 2.5 Pro achieved the highest

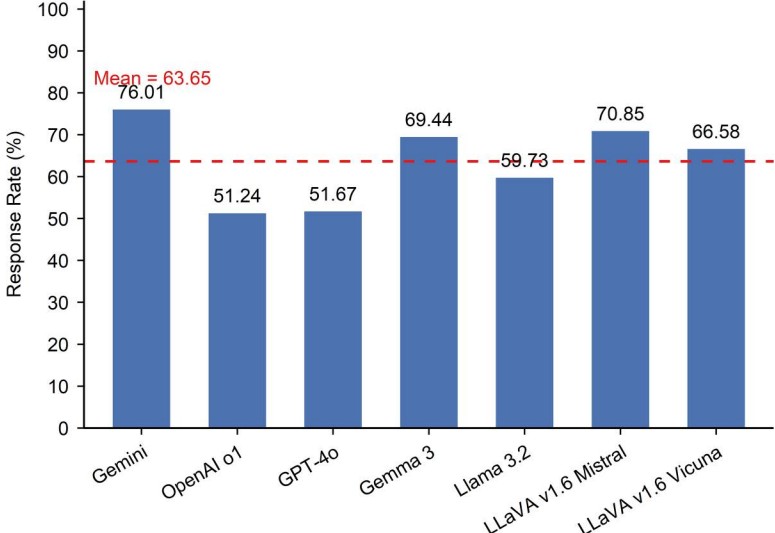

**Fig 2. Overall response rates of MLLMs across all task categories.**

rate at 76.01%, demonstrating strong responsiveness to task prompts. In contrast, OpenAI o1 and GPT-4o recorded the lowest rates, at 51.24% and 51.67%, respectively. Among the open-source models, LLaVA v1.6 Mistral, Gemma 3, and LLaVA v1.6 Vicuna achieved relatively high response rates of 70.85%, 69.44%, and 66.58%, respectively. LlaMA 3.2 had the lowest response rate within this group, at 59.73%, slightly below the overall average.

Table 2 reports model response rates by the four task categories. Overall, MLLMs responded well to prompts focused on pavement surface distress and feature identification (71.82%), pavement condition description and severity evaluation (77.86%), and maintenance interval estimation (94.86%). However, the average response rate dropped significantly for spatial pattern recognition tasks (47.06%). These findings suggest that current MLLMs are generally capable of responding to tasks involving object identification and text-based reasoning, such as describing pavement distress and estimating maintenance needs. However, they struggle with tasks that require interpreting visual information related to spatial distribution patterns.

Pavement surface distress and feature identification exhibited varied response rates across models. Gemini 2.5 Pro (88.30%), LLaVA v1.6 Mistral (81.52%), and Gemma 3 (78.72%) showed higher overall response rates for pavement surface distress and feature identification, whereas OpenAI o1 (52.05%) and GPT-4o (53.49%) performed lower. Reviewing

**Table 2. Average response rates (%) by task category.**

| MLLMs | Pavement surface distress and feature identification | Spatial pattern recognition | Pavement condition description and severity evaluation | Maintenance interval estimation |
|---|---|---|---|---|
| Gemini 2.5 Pro | 88.30 | 57.14 | 89.87 | 93.49 |
| OpenAI o1 | 52.05 | 29.58 | 83.53 | 96.93 |
| GPT-4o | 53.49 | 28.98 | 84.20 | 98.47 |
| Gemma 3 | 78.72 | 48.61 | 89.66 | 100.00 |
| Llama 3.2 | 70.32 | 47.39 | 62.50 | 94.64 |
| LLaVA v1.6 Mistral | 81.52 | 59.24 | 73.43 | 94.64 |
| LLaVA v1.6 Vicuna | 78.34 | 58.48 | 61.86 | 85.82 |
| Overall | 71.82 | 47.06 | 77.86 | 94.86 |

the subtask-level results in Table 3, most models responded well to general questions, such as identifying whether cracks are present, with response rates close to or at 100% (e.g., 99.62% for OpenAI o1 and 100.00% for GPT-4o). However, their responsiveness dropped markedly for fine-grained classification tasks. For example, in the subtask of identifying transverse cracks, the response rates of OpenAI o1 and GPT-4o dropped to 42.91% and 44.06%, respectively. These findings suggest that the lower overall responsiveness of models like OpenAI o1 and GPT-4o is largely due to their limited performance on fine-grained tasks involving specific distress types.

Spatial pattern recognition had the lowest response rates among all tasks, with GPT-4o performing the weakest at just 28.98%, and Gemini 2.5 Pro achieving the highest at 57.14%. These results indicate that most models struggle with solving tasks related to the spatial distribution, patterns, or layout of road damage. Compared to proprietary models, open-source models generally outperformed proprietary models in spatial pattern recognition. LLaVA v1.6 Mistral (59.24%), LLaVA v1.6 Vicuna (58.48%), and Gemma 3 (48.61%) achieved higher response rates than GPT-4o (28.98%) and OpenAI o1 (29.58%), suggesting that open-source architecture may be better suited for spatial reasoning tasks. Nevertheless, all models showed markedly lower response rates in spatial pattern recognition than in other tasks, underscoring the persistent difficulty of capturing spatial context and relational features in road surface imagery. Pavement condition description and severity evaluation presented strong response rates across models, with an overall average of 77.86%. Most models were capable of generating concise textual descriptions of the road environment and estimating damage severity. Gemini 2.5 Pro (89.87%) and Gemma 3 (89.66%) achieved the highest response rates in this task category, while LlaMA 3.2 recorded the lowest at 62.50%.

Maintenance interval estimation had the highest response rates among all task categories, with an overall average of 94.86%. Gemma 3 reached 100%, followed by GPT-4o (98.47%), OpenAI o1 (96.93%), and Gemini 2.5 Pro (93.49%). The remaining models also performed well, each with response rates above 85%. These results indicate that most MLLMs are able to generate outputs when prompted to estimate appropriate maintenance timing based on road condition inputs.

## Response correctness

Response correctness varies across models, with proprietary models generally outperforming their open-source counterparts. Fig 3 compares the response correctness of seven MLLMs across four metrics: (a) accuracy, (b) precision, (c) recall, and (d) F1 score. GPT-4o and OpenAI o1 (proprietary models) achieved the highest scores across all metrics,

**Table 3. Response rates (%) of MLLMs by subtask in pavement surface distress and feature identification.**

| MLLMs | Cracks | | | | | Potholes | | Other infrastructure | | Pavement Surface Treatments | | | |
|---|---|---|---|---|---|---|---|---|---|---|---|---|---|
| | Cracks in general | Trans-verse cracks | Longi-tudinal cracks | Alli-gator cracks | Other types of cracks | Pot-holes present | Number of pot-holes | Utility cuts or joints | Drain-age | Repair-ment in general | Patched areas | Sealed cracks | Resur-facing |
| Gemini 2.5 Pro | 94.25 | 80.84 | 80.84 | 80.84 | 77.78 | 94.25 | 90.80 | 94.25 | 89.66 | 92.34 | 91.95 | 90.80 | 89.27 |
| OpenAI o1 | 99.62 | 42.91 | 42.91 | 42.91 | 41.76 | 99.62 | 45.21 | 99.23 | 28.35 | 36.78 | 36.78 | 36.78 | 23.75 |
| GPT-4o | 100.00 | 44.06 | 44.06 | 44.06 | 41.76 | 100.00 | 44.06 | 98.85 | 28.74 | 41.76 | 41.76 | 41.76 | 24.52 |
| Gemma 3 | 100.00 | 83.14 | 83.14 | 83.14 | 82.76 | 100.00 | 34.48 | 99.62 | 70.11 | 74.33 | 73.95 | 73.95 | 64.75 |
| Llama 3.2 | 99.62 | 90.80 | 90.80 | 90.80 | 73.56 | 99.62 | 89.66 | 83.14 | 2.30 | 63.98 | 63.98 | 63.98 | 1.92 |
| LLaVA v1.6 Mistral | 100.00 | 99.62 | 99.62 | 99.62 | 78.93 | 99.62 | 90.42 | 95.79 | 26.82 | 81.99 | 81.61 | 81.23 | 24.52 |
| LLaVA v1.6 Vicuna | 100.00 | 99.62 | 98.85 | 98.47 | 42.91 | 100.00 | 98.85 | 96.93 | 1.53 | 93.49 | 93.87 | 93.49 | 0.38 |
| Overall | 99.07 | 77.28 | 77.17 | 77.12 | 62.78 | 99.02 | 70.50 | 95.40 | 35.36 | 69.24 | 69.13 | 68.86 | 32.73 |

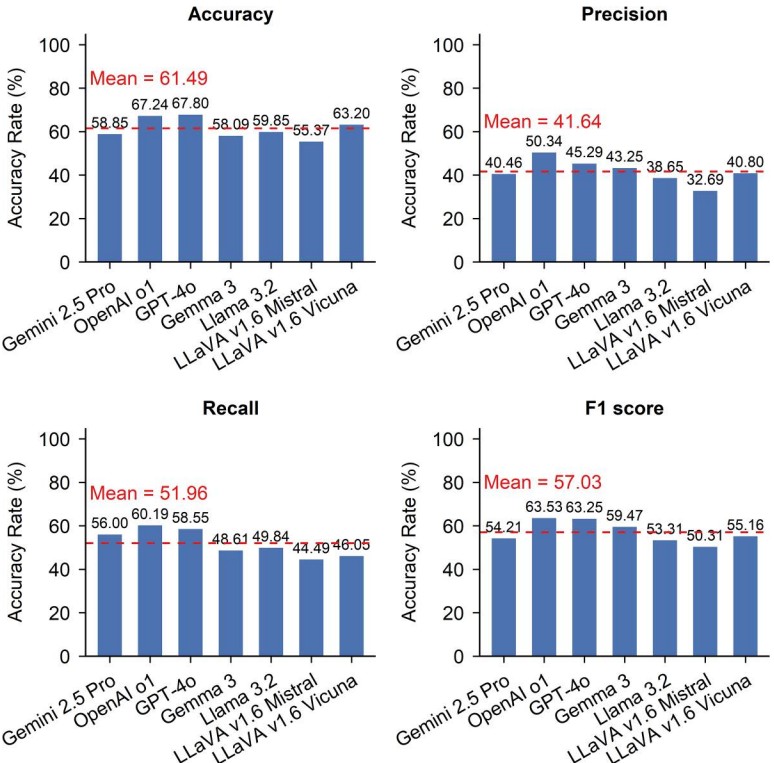

**Fig 3. Overall response correctness of MLLMs across all task categories.**

indicating stronger performance in generating accurate, relevant, and complete outputs for road condition assessment tasks. The red dashed line represents the mean score for that metric. In contrast, open-source models consistently underperformed, particularly in precision and recall, suggesting a higher likelihood of producing false positives and omitting relevant features.

As shown in Fig 3(a), GPT-4o achieved the highest accuracy at 67.80%, followed closely by OpenAI o1 at 67.24%. LLaVA v1.6 Vicuna was the only open-source model to exceed the group average of 61.49%, achieving an accuracy of 63.20%. Other models such as Gemini 2.5 Pro (58.85%), Gemma 3 (58.09%), and LLaVA v1.6 Mistral (55.37%) fell below the average, indicating more frequent generation of incorrect outputs. These results suggest that GPT-based models are more effective at producing correct responses across road condition assessment tasks.

In terms of precision, shown in Fig 3(b), OpenAI o1 led with 50.34%, followed by GPT-4o at 45.29%. Although these values were the highest among the evaluated models, they indicate that even the top-performing models occasionally labeled incorrect outputs as correct. Precision values for Gemma 3, LLaVA v1.6 Mistral, and LLaVA v1.6 Vicuna ranged between 32.69% and 43.25%, suggesting a greater tendency to produce false positives. Gemini 2.5 Pro recorded a precision of 40.46%, which was below the group mean of 41.64%.

Recall scores, displayed in Fig 3(c), followed a similar pattern to accuracy and precision. OpenAI o1 (60.19%), GPT-4o (58.55%), and Gemini 2.5 Pro (56.00%) all outperformed the group average of 51.96%, indicating they were better at correctly identifying relevant information for surface road condition assessment tasks. In contrast, all open-source models had lower recall scores, ranging from 44.49% to 49.84%, suggesting they were more likely to miss important features during the assessment.

Fig 3(d) presents F1 scores, which balance precision and recall providing an overall measure of response correctness. OpenAI o1 and GPT-4o again led this metric, with scores of 63.53% and 63.25%, respectively. Gemma 3 achieved a moderately strong score of 59.47%, while Gemini 2.5 Pro lagged behind at 54.21% due to lower precision. Open-source models including LLaVA v1.6 Mistral (50.31%), LLaVA v1.6 Vicuna (55.16%), and Llama 3.2 (53.34%) all scored below the group mean of 57.03%. These results confirm that proprietary models are better equipped to deliver responses for the assessment.

Table 4 presents response correctness metrics across the four task categories. Results show that models performed better on visual recognition tasks, such as pavement surface distress and feature identification as well as spatial pattern recognition, but showed weaker performance on interpretive reasoning tasks, including pavement condition evaluation and maintenance interval estimation.

Among the visual recognition tasks, the highest performance was observed in pavement surface distress and feature identification. GPT-4o (Accuracy: 78.99%, F1: 69.50%) and OpenAI o1 (Accuracy: 77.55%, F1: 67.72%) demonstrated strong capabilities in detecting visible surface conditions such as cracks and potholes. Similarly, in spatial pattern recognition, both models maintained relatively high performance, with OpenAI o1 achieving an F1 score of 68.53% and GPT-4o scoring 66.55%.

In contrast, interpretive reasoning tasks are more challenging, as they require models to infer severity levels or predict future repair actions based on complex and often ambiguous visual and contextual cues. Consequently, model performance in these tasks remains low. All models recorded F1 scores below 25% for pavement condition evaluation and below 35% for maintenance interval estimation, highlighting their limited capability in producing accurate and consistent predictions.

## Consistency

Fig 4 shows that proprietary models tend to produce more consistent outputs, and open-source models generate more variable and less predictable responses. Specifically, proprietary models exhibited stronger inter-model agreement than open-source models. The highest pairwise agreement was observed between GPT-4o and OpenAI o1 (0.621), indicating a high level of consistency in their outputs, likely due to similar underlying architectures or training strategies. Gemini 2.5 Pro also showed moderate agreement with GPT-4o (0.181) and OpenAI o1 (0.196), further supporting the coherence within proprietary models. In contrast, open-source models demonstrated low or near-zero agreement both among themselves and with proprietary models. For example, LLaVA v1.6 Vicuna showed almost no agreement with other models,

**Table 4. Response correctness metrics (%) by task category.**

| MLLMs | Pavement surface distress and feature identification | | | | Spatial pattern recognition | | | | Pavement condition evaluation | | | | Maintenance interval estimation | | | |
|---|---|---|---|---|---|---|---|---|---|---|---|---|---|---|---|---|
| | Accuracy | Precision | Recall | F1 score | Accuracy | Precision | Recall | F1 score | Accuracy | Precision | Recall | F1 score | Accuracy | Precision | Recall | F1 score |
| Gemini 2.5 Pro | 73.04 | 49.49 | 65.35 | 64.85 | 56.43 | 37.37 | 54.98 | 52.06 | 17.50 | 24.00 | 18.20 | 22.30 | 23.00 | 30.50 | 40.90 | 21.90 |
| OpenAI o1 | 77.55 | 60.65 | 69.21 | 67.72 | 67.16 | 48.68 | 60.02 | 68.53 | 19.85 | 19.10 | 17.65 | 20.85 | 44.70 | 29.50 | 40.10 | 32.70 |
| GPT-4o | 78.99 | 55.32 | 66.79 | 69.50 | 67.11 | 41.69 | 58.50 | 66.55 | 19.80 | 25.30 | 19.45 | 19.80 | 45.90 | 22.00 | 40.80 | 33.30 |
| Gemma 3 | 68.21 | 38.53 | 58.48 | 66.46 | 62.63 | 56.55 | 49.63 | 65.40 | 13.20 | 18.30 | 11.95 | 13.25 | 3.10 | 18.20 | 23.10 | 5.60 |
| Llama 3.2 | 67.41 | 44.70 | 55.81 | 57.33 | 63.99 | 39.68 | 52.88 | 58.93 | 14.65 | 17.60 | 12.05 | 19.65 | 9.30 | 17.80 | 22.10 | 13.20 |
| LLaVA v1.6 Mistral | 57.25 | 34.05 | 48.08 | 50.17 | 65.44 | 36.49 | 47.91 | 60.07 | 11.45 | 12.90 | 12.60 | 15.45 | 7.70 | 13.20 | 25.90 | 9.20 |
| LLaVA v1.6 Vicuna | 68.74 | 45.85 | 52.38 | 58.44 | 68.47 | 41.79 | 47.86 | 60.58 | 18.40 | 10.40 | 10.80 | 23.35 | 24.60 | 22.60 | 20.60 | 27.30 |
| Overall | 70.17 | 46.94 | 59.44 | 62.07 | 64.46 | 43.18 | 53.11 | 61.73 | 16.41 | 18.23 | 14.67 | 19.24 | 22.61 | 21.97 | 30.50 | 20.46 |

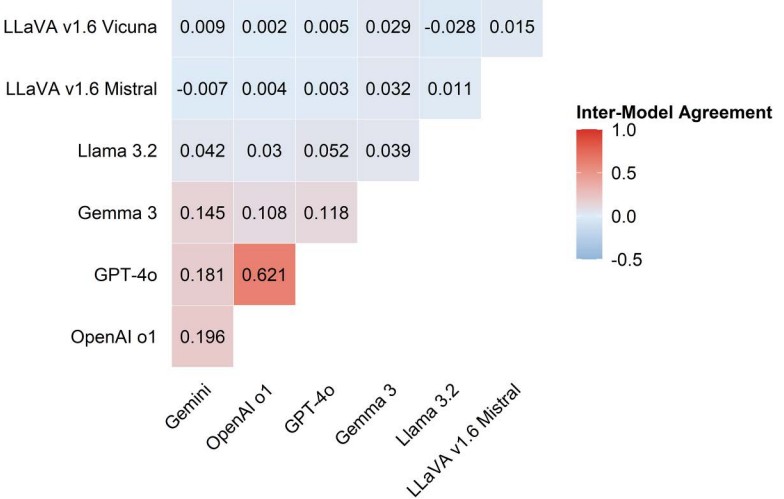

**Fig 4. Average inter-model agreement between MLLMs.**

with values ranging from 0.002 to 0.029, and even a slightly negative agreement with LLaVA v1.6 Mistral (−0.028). Similarly, LLaVA v1.6 Mistral and Llama 3.2 showed weak inter-model alignment, suggesting inconsistent output behavior.

Fig 5 presents inter-model agreements across four task categories. Each heatmap panel shows agreement on a road condition assessment task: (a) pavement surface distress and feature identification, (b) spatial pattern Recognition, (c) road condition description and severity Evaluation, and (d) maintenance interval estimation. GPT-4o and OpenAI o1 showed high agreement in all four task categories, with particularly strong alignment in pavement surface distress identification and damage pattern recognition. This suggests that GPT-based models are more consistent in visual recognition tasks. Gemini 2.5 Pro showed moderate agreement with both GPT-4o and OpenAI o1 across all tasks, indicating relatively stable response behavior and internal consistency among proprietary models. In contrast, open-source models showed low agreement both among themselves and with proprietary models across all task categories. For example, LLaVA v1.6 Vicuna and Mistral frequently yielded near-zero or even negative agreement scores, indicating that they often produced divergent outputs when processing the same input. Such inconsistency suggests that open-source MLLMs may be less dependable for road condition assessment tasks, particularly those that require consistent recognition and interpretation of visual and contextual information.

## Multimodal errors

**Pavement surface distress and feature identification.** MLLMs often struggle to identify specific types of surface distress when performing fine-grained classification. While most models reliably detected the general presence of cracks or potholes, they frequently misidentified the specific type of distress. In several instances, models hallucinate cracks type, such as labeling transverse cracks as longitudinal or falsely detected alligator cracks in unrelated patterns. These hallucinations commonly occurred when distress features were subtle, partially obscured, or visually similar, making it difficult for the model to apply precise visual reasoning. This pattern suggests that although object detection is often accurate at a general level, the step of assigning a detailed label is prone to error. Such misclassifications introduce risks when outputs are used to inform infrastructure repair decisions, particularly in workflows where the type of surface distress directly affects prioritization or treatment strategy.

**Spatial pattern recognition.** MLLMs often encounter difficulty when interpreting how surface damage is distributed across roadways, particularly in tasks that require recognizing spatial layouts or structural alignment. Unlike general

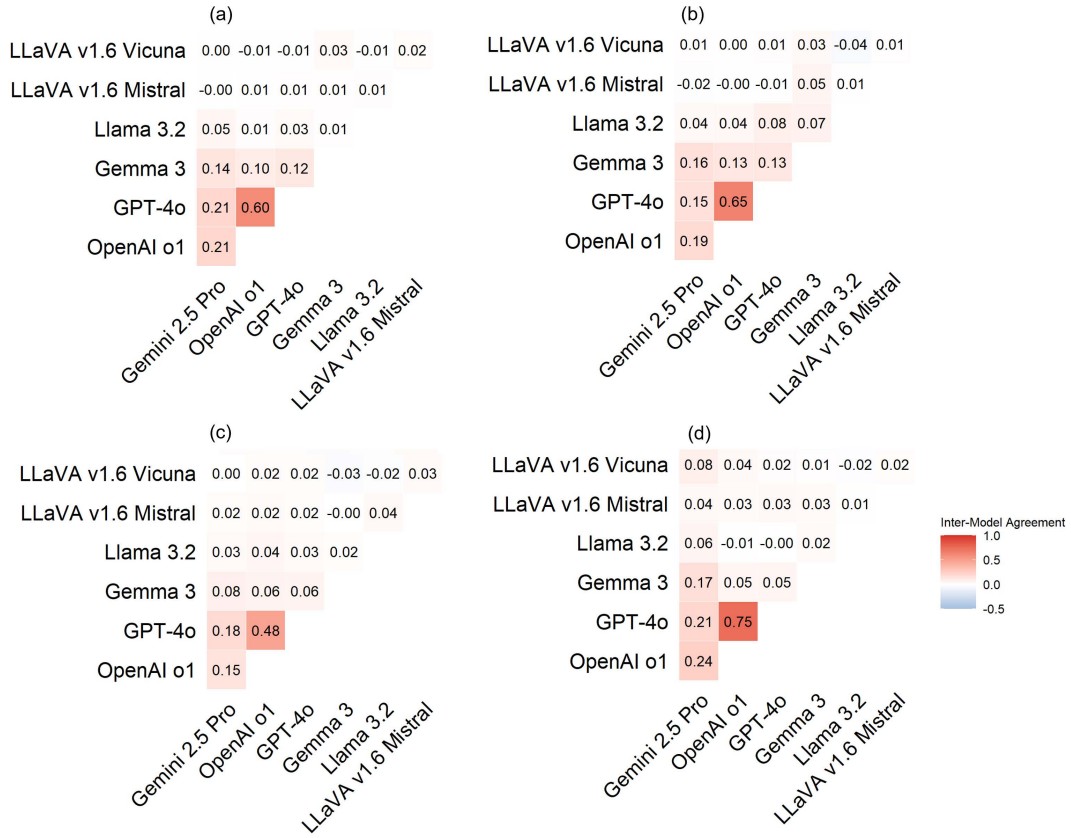

**Fig 5. Pairwise inter-model agreement across MLLMs.**

damage detection, spatial pattern recognition demands a more nuanced understanding of how cracks or potholes relate to features such as joints, edges, or intersections. In these cases, models frequently misclassified or hallucinated spatial arrangements. These errors were especially common when features were irregular, subtle, or embedded in complex visual contexts. Rather than relying solely on observed visual cues, models may infer complexity where none exists, leading to exaggerated or inaccurate spatial descriptions. This poses a challenge for applications that require accurate diagnosis of localized deterioration, such as planning targeted maintenance interventions.

**Pavement severity assessment.** Discrepancies were observed between MLLM-generated severity assessments and PCI scores provided by the City of San Francisco. These differences do not necessarily indicate model error, as the PCI scores are based on human evaluations that may introduce subjectivity. As shown in S2 Table, in some cases, MLLM assessments more accurately reflect the visible surface conditions captured in the street view images. This suggests that PCI labels may not always provide the most up-to-date or visually consistent representation of roadway quality, and that MLLMs can offer a valid alternative based on visual evidence. At the same time, it is important to note that in this study, the prompt requested only a PCI value without requiring justification. Without an explicit requirement for reasoning, the model may generate estimated values based on assumptions or unsupported visual cues.

**Pavement maintenance interval estimation.** Multimodal errors in responses to maintenance interval estimation occurred when models relied excessively on visible surface distress, leading them to suggest short-term repairs for roads that were not actually scheduled for maintenance. This may have happened because important contextual factors such as long-term transportation plans, budget limitations, and traffic management needs were not considered, as shown in S3

Table. Although the road conditions appeared to be deteriorated, real-world maintenance decisions often prioritize these broader considerations, leading to delayed repairs. This behavior could represent a form of hallucination in which the model inferred unwarranted maintenance needs based solely on surface appearance, generating recommendations that were not grounded in actual maintenance schedules or operational priorities.

## Overall computational intensity and cost

Fig 6 compares the processing time (left) and token count (right) across all evaluated MLLMs. Red lines indicate median values, annotated above each box. Proprietary models exhibited longer processing times per street image. OpenAI o1 and GPT-4o had median durations of 67.25 and 52.08 seconds, respectively, while Gemini 2.5 Pro was faster at 40.79 seconds. In contrast, open-source models showed consistently shorter processing times with minimal variability.

Token usage varied by model. Among the proprietary models, Gemini 2.5 Pro had the highest median token usage (3,445), followed by GPT-4o (3,304) and OpenAI o1 (3,196). Interestingly, with input prompts and image embeddings kept consistent across cases, the near-constant token usage of open-source models suggests a fixed-length output behavior. This pattern suggests that open-source models may rely more heavily on templated responses, while proprietary models adjust output length dynamically based on input content.

A Kruskal-Wallis test revealed significant differences in token usage among the three models ($p = 0.0066$), with a small effect size ($\eta^2 = 0.010$). Post hoc pairwise Wilcoxon rank-sum tests with Bonferroni correction showed that Gemini 2.5 Pro used a significantly different number of tokens compared to both GPT-4o (adjusted $p = 0.027$, $r = 0.11$) and OpenAI o1 (adjusted $p = 0.018$, $r = 0.12$), while no significant difference was found between GPT-4o and OpenAI o1 (adjusted

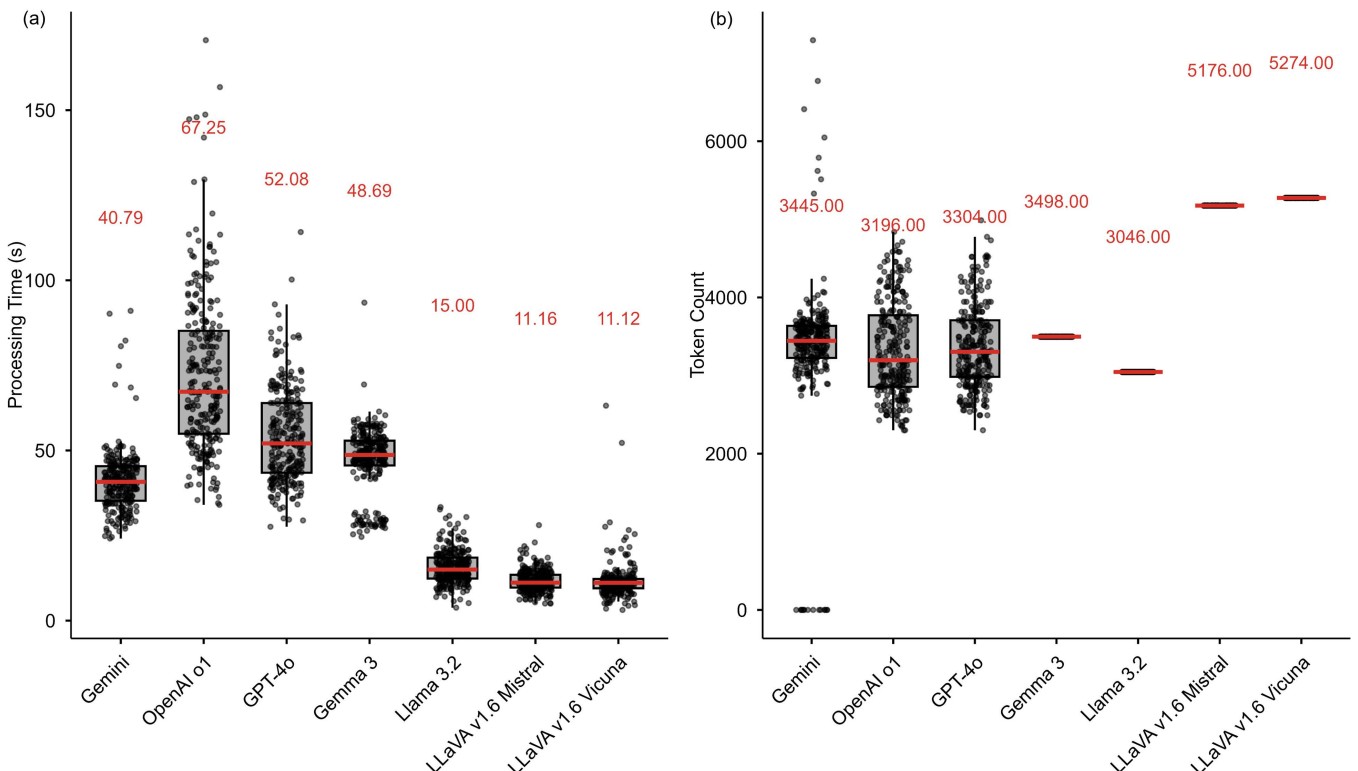

**Fig 6. Comparison of processing time and token count across MLLMs.**

p = 1.000, r = 0.04). Despite statistical significance, the small effect sizes suggest that these differences in token usage may have limited practical relevance on their own.

Translating token usage into monetary cost, analyzing one street image with 39 prompts related to road surface condition assessment costs approximately $0.02 using Gemini 2.5 Pro, $0.04 using GPT-4o, and $0.12 using OpenAI o1. Although Gemini 2.5 Pro typically consumes more tokens, its lower per-token rate results in a lower overall cost compared to OpenAI o1. The per-token cost of OpenAI o1 is approximately seven times higher than that of Gemini 2.5 Pro and at least 3.75 times higher than that of GPT-4o. Consequently, OpenAI o1 incurs the highest cost per image, while Gemini 2.5 Pro and GPT-4o offer more cost-efficient alternatives.

## Discussion, limitations and future work

### The potential and current limitations of existing MLLMs in road surface condition assessment

The study assessed the performance of both proprietary and open-source MLLMs on four tasks relevant to surface road condition assessment. Results indicate that these models can analyze street-level imagery and generate relevant, structured responses that align with task objectives. This confirmed the potential of MLLMs to support surface road condition assessment workflows.

Beyond performance, MLLMs are characterized by their user-friendly design and low operational costs, which support their applicability in pavement condition assessment. The user-friendliness of MLLMs is reflected in two aspects. First, they allow users to interact with the model using natural language prompts [83], removing the need for specialized programming expertise. Second, MLLMs can produce structured assessment outputs directly from raw street-level images, without requiring extensive data preprocessing or formatting. This streamlined input process reduces the technical burden typically associated with automated assessment tools and facilitates more efficient deployment in real-world pavement evaluation workflows.

In terms of cost, MLLMs are also relatively affordable. The cost of analysis ranges from $0.02 to $0.12 per image. In contrast, conventional road condition assessments can be significantly more expensive, with costs starting at approximately $3,772 for evaluating a roadway of at least one mile in length [84]. Assuming one image is captured every 25 feet, or approximately 212 images per mile, an MLLM-based assessment would cost between $4.24 and $25.44 for a one-mile segment.

However, these advantages in usability and affordability are accompanied by several limitations. MLLM performance is not consistent across all task types and tends to vary with task complexity. The models perform well on general tasks that require minimal contextual understanding, such as identifying the presence of cracks or potholes. Their performance weakens as tasks require more detailed or specific classifications. For example, the average response rate drops from 99.07% when identifying general cracks to 77.28% when identifying transverse cracks. Accuracy also declines, with the average F1 score decreasing from 0.62 for object identification to 0.20 for maintenance interval estimation.

Another factor affecting model performance is the clarity and completeness of the input prompt. Models often produce inaccurate responses when the input lacks sufficient context. For instance, in this study, the prompt only asked the models to recommend maintenance intervals based on street-view images, without providing any additional background information. As a result, the models focused exclusively on visible pavement conditions and overlooked non-visual determinants of maintenance needs. Incorporating these contextual factors into model prompts or fine-tuning datasets could improve both the reliability and practical utility of MLLM-generated outputs.

Furthermore, generative models may produce probabilistic outputs, meaning that repeated runs do not necessarily yield identical results unless generation parameters are fixed. This stochastic variability parallels observer inconsistency in manual inspections and can introduce uncertainty into the evaluation process. Standardizing model parameters and employing well-structured prompts can mitigate this randomness, thereby enhancing the reproducibility and reliability of MLLM outputs.

A further limitation lies in the explainability of MLLM outputs. Although the models can generate decisions related to surface road condition assessment, their reasoning processes remain opaque. For instance, when asked to provide a PCI based on an image, the models typically return only a numerical value without clarifying how it was derived. While MLLMs can produce descriptive explanations, these are not always grounded in verifiable causal relationships and may reflect post-hoc rationalizations rather than genuine interpretive reasoning. Prompting models to explicitly state the rationale behind their decisions could improve transparency and facilitate more trustworthy applications in pavement assessment.

Finally, Although GAI can reduce reliance on manual inspections by automating visual assessments, human oversight remains necessary to validate cases and ensure alignment with engineering standards. While GAI using street-view imagery can assist with tasks related to surface condition assessment, effective pavement management requires additional data beyond surface imagery such as structural integrity and traffic loading. Therefore, GAI should be viewed as a decision-support toolkit that enhances human judgment rather than a complete replacement for professional expertise and field validation.

## Pathways to improving MLLM performance

Enhancing their effectiveness on road surface condition assessment will require improvements in several areas. One key area is the need for additional domain-specific training data to support fine-tuning and improve model performance [85]. Many core assessment tasks, such as identifying specific types of pavement cracks or determining what types of repairs have been made to the road (e.g., patched areas, resurfacing), require detailed recognition capabilities that current models often lack. Compared to general object detection, the response rate for these fine-grained tasks is significantly lower, indicating that MLLMs need further tuning to recognize subtle distinctions in road surface conditions. Similarly, tasks such as pavement severity estimation and maintenance interval prediction depend on contextual variables beyond the visible defect itself. These tasks require reasoning based on factors such as budget, environmental exposure, or traffic patterns, which are typically not part of the model's default knowledge. Fine-tuning with annotated datasets that include these features is necessary to improve model performance on these complex tasks.

In parallel, knowledge distillation can serve as a complementary strategy [86]. By transferring knowledge from high-performing proprietary models to open-source models, distillation can enhance performance without the computational cost of training from scratch. This is particularly useful for tasks like spatial pattern recognition, where response rates may be low, but the accuracy of responses, when provided, is relatively high. This suggests that models possess some capacity for spatial reasoning, but they are not consistently activated. Distillation can help encode this reasoning more effectively, enabling smaller models to perform better in both classification and spatial interpretation tasks.

At the same time, prompt engineering plays an important role in improving model outputs, particularly for context-dependent tasks [87]. For instance, when asking when a road should be repaired, the model must consider information not visible in the image, such as usage intensity, historical repair records, or surrounding infrastructure. Designing prompts that incorporate this contextual information and guide the model to consider relevant variables can significantly improve response quality. This approach is especially valuable in situations where retraining or fine-tuning is not feasible, as it allows users to optimize model performance through carefully structured input design.

## Proprietary vs. open-source models: trade-offs and implications for future development

This study compared the performance of proprietary and open-source MLLMs across multiple dimensions, including responsiveness, accuracy, internal agreement, and cost. While both categories of models demonstrate similar levels of responsiveness, proprietary models generally achieve higher accuracy and show stronger internal agreement, producing similar outputs when presented with the same or slightly modified prompts. These advantages make proprietary models highly reliable for operational deployment; however, their use involves financial costs and limited transparency. Most

proprietary models cannot be fine-tuned or adapted for domain-specific applications, and their token-based pricing structures can substantially increase computational expenses, especially in large-scale analyses.

In contrast, open-source models are freely available and offer greater flexibility for customization and domain adaptation. Although they typically perform slightly worse than proprietary models in accuracy and consistency, their transparency, accessibility, and modifiability make them valuable for research and development. Open-source models can be fine-tuned on task-specific datasets, enabling iterative improvement without recurring costs. This balance between performance and accessibility suggests that open-source models are a practical foundation for future pavement assessment applications, particularly in settings where reproducibility and budget constraints are critical.

Among the open-source models evaluated, Gemma 3 is recommended as the base model for fine-tuning. It demonstrates high response rates across all tasks, including a perfect score (100%) for maintenance interval estimation. Gemma 3 also achieves the highest F1 score among all open-source models, exceedingly even that of Gemini 2.5 Pro, which indicates strong overall reliability and balanced performance. Furthermore, Gemma 3 shows higher consistency with proprietary models compared to other open-source models. In contrast, the relatively low consistency observed among the LLaVA and Llama models suggests that their responses may be more variable or randomly generated. Overall, Gemma 3 achieved the best performance across all tasks, suggesting it is a strong candidate for fine-tuning and domain-specific adaptation. For applications involving knowledge distillation, GPT-4o is the most suitable proprietary model. It achieves high accuracy and internal consistency, comparable to OpenAI o1, but at a lower operational cost.

## Conclusion

In this study, we evaluated the performance of seven MLLMs, including three proprietary models (GPT-4o, OpenAI o1, Gemini 2.5 Pro) and four open-source models (LLaVA v1.6 Mistral, LLaVA v1.6 Vicuna, Llama 3.2, Gemma 3), across four task categories: (1) pavement surface distress and feature identification, (2) spatial pattern recognition, (3) pavement condition severity evaluation, and (4) pavement maintenance interval estimation. The models were assessed across five dimensions: response rate, accuracy, consistency, multimodal errors, and computational cost.

The results empirically verified the transformative potential of MLLMs, which lies in their strong responsiveness and low cost. On average, the models achieved a 63.65% response rate, and the cost of conducting an assessment based on a single image ranged from approximately $0.02 to $0.12, depending on the model used. While accuracy requires further optimization, this work establishes a benchmark proving that with domain-specific refinement, MLLMs can transition from assistive tools to scalable, cost-effective solutions for pavement monitoring and management.

## Supporting information

**S1 Table. List of prompts used for model evaluation.**
(DOCX)

**S2 Table. Cases illustrate discrepancies between MLLM-generated severity assessments and PCI-based condition label.**
(DOCX)

**S3 Table. The comparison between repair maintenance intervals estimated by MLLMs and actual intervals.**
(DOCX)

## Author contributions

**Conceptualization:** Chang Xu, Lei Shu, Yue Cui.

**Data curation:** Chang Xu, Anh Dao.

**Formal analysis:** Chang Xu, Lei Shu.

**Funding acquisition:** Yue Cui.

**Investigation:** Chang Xu, Yue Cui.

**Methodology:** Chang Xu, Lei Shu, Anh Dao, Yue Cui.

**Project administration:** Yue Cui.

**Resources:** Yue Cui.

**Software:** Anh Dao, Yue Cui.

**Supervision:** Yue Cui.

**Validation:** Chang Xu, Lei Shu.

**Visualization:** Chang Xu.

**Writing – original draft:** Chang Xu, Lei Shu, Yue Cui.

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
