## [Decision Letter · Decision Letter 0]

6 Oct 2025

PONE-D-25-46997Multimodal Generative AI for Automated Pavement Condition Assessment: Benchmarking Model PerformancePLOS ONE

Dear Dr. Cui,

Thank you for submitting your manuscript to PLOS ONE. After careful consideration, we feel that it has merit but does not fully meet PLOS ONE’s publication criteria as it currently stands. Therefore, we invite you to submit a revised version of the manuscript that addresses the points raised during the review process.

We look forward to receiving your revised manuscript.

Kind regards,

Junghwan Kim

Academic Editor

PLOS ONE

Journal Requirements:

“CESU grant

Funding Opportunity No: W912HZ-19-SOI-0002”

4. In the online submission form, you indicated that your data will be submitted to a repository upon acceptance.  We strongly recommend all authors deposit their data before acceptance, as the process can be lengthy and hold up publication timelines. Please note that, though access restrictions are acceptable now, your entire minimal  dataset will need to be made freely accessible if your manuscript is accepted for publication. This policy applies to all data except where public deposition would breach compliance with the protocol approved by your research ethics board. If you are unable to adhere to our open data policy, please kindly revise your statement to explain your reasoning and we will seek the editor's input on an exemption.

Additional Editor Comments (if provided):

Please ensure to address the reviewers' comments thoroughly. Additionally, please expand the literature review of recent papers on this topic to discuss gaps in the existing studies and why those gaps are significant. There are many papers on the similar or relevant topics (i.e., LLM + Street-view images) published in geography and urban planning journals, which have not been extensively discussed in this manuscript.

Reviewers' comments:

Reviewer's Responses to Questions

**Comments to the Author**

1. Is the manuscript technically sound, and do the data support the conclusions?

Reviewer #1: Partly

Reviewer #2: Yes

Reviewer #3: Yes

2. Has the statistical analysis been performed appropriately and rigorously? 

Reviewer #1: No

Reviewer #2: Yes

Reviewer #3: Yes

3. Have the authors made all data underlying the findings in their manuscript fully available?

Reviewer #1: No

Reviewer #2: No

Reviewer #3: Yes

4. Is the manuscript presented in an intelligible fashion and written in standard English?

Reviewer #1: Yes

Reviewer #2: Yes

Reviewer #3: Yes

5. Review Comments to the Author

Reviewer #1: This paper proposes using large language models (LLMs) to audit pavement condition. I think many technical details need to be described and considered more carefully.

Abstract

1. Page 1: The abstract starts by saying LLMs show promise in automated assessment of many things, but automated pavement condition assessment is unexplored. I don’t think this is a strong motivation (there are countless unexplored applications of LLMs). Instead, the authors should start with current challenges in pavement condition assessment and then explain how LLMs could help address these challenges.

Introduction

2. Page 2 (lines 36-39): Proper references should be cited for existing work on “structural evaluation” and “surface-level evaluation.” The authors should also specify which physical characteristics these studies focus on, with a few examples.

3. Page 3 (lines 46-56): Several claims here seem factually inaccurate. First, when you say existing deep learning models require “high-quality, structured image data,” do you mean training data? Even for LLMs, you still need high-quality image data to support detection and decision-making. Second, while deep learning models need human oversight, so do LLMs, which can hallucinate and require human review (so they can’t truly autonomously make decisions). Perhaps the authors mean that a single LLM could potentially handle not just detection but also evaluation of severity and maintenance timing, tasks that would require multiple deep learning models.

Methods

4. Page 7: First, what are the 39 prompts used? These should be spelled out either here or in an appendix, as it is currently unclear what the authors are asking the LLMs to do. Second, how are the model parameters set, such as temperature, top-k, etc.? Third, for each LLM, is it only prompted once per street view image? Typically, to enhance robustness in LLM-assisted assessment, each LLM should be prompted multiple times per image, because responses can vary due to inherent randomness.

5. Page 8: More details are needed about the manual annotation process. For example, where were the annotators recruited? Provide a detailed list of labels used for surface features and spatial distribution patterns, not just examples. The examples given here could instead go in the Introduction (page 3, lines 57–63), because terms like “spatial distribution pattern” are unclear until this point.

6. Page 11: How do you distinguish between response correctness and hallucination in practice? Both result in a mismatch between human-annotated data and LLM responses.

7. Echoing my earlier points, the authors should be explicit about what they instruct the LLMs to do, what outputs are expected, and what is contained in the human-annotated data. Without this clarity, it is difficult to understand how performance evaluation is conducted. How do we know whether the human-annotated results and LLM responses are comparable?

Reviewer #2: The manuscript titled “Multimodal Generative AI for Automated Pavement Condition Assessment: Benchmarking Model Performance” addresses an important and timely problem by exploring the application of multimodal generative AI to pavement condition assessment. The study provides a clear comparative evaluation of multiple proprietary and open-source models and presents results that highlight relative performance differences across tasks and dimensions. Here are my comments.

- While the study demonstrates that multimodal generative AI models can produce pavement maintenance recommendations, one important limitation not addressed in the manuscript is the lack of explainability of model outputs. For infrastructure management such as pavement condition assessment, where decisions directly affect safety, costs, and policy priorities, explainability is critical. Without interpretable reasoning or transparent links between input features (e.g., crack severity, traffic data) and recommendations, practitioners may be reluctant to trust or adopt such AI systems. The authors should discuss how LLMs address this.

- As one of the limitations of manual inspections, the paper noted that manual inspections are subject to observer variability (ln 40-41). This is also applicable in utilizing generative AI too. Many models generate probabilistic outputs, meaning repeated runs may not always yield identical results unless parameters are fixed. This needs to be acknowledged in this study

- For the 39 prompts used in this study, are there any domain expertise involved in their design? Given that prompt clarity and completeness strongly affect model outputs, it will benefit this paper to clarify whether experts in pavement management were consulted, and what criteria guided prompt construction.

- Response rate was used as one of the assessment dimensions. The paper does not clearly explain how response rate was measured in practice. Was it calculated manually?

Reviewer #3: The manuscript presents a timely contribution to the growing field of using MLLMs for imagery analysis. The integration of GenAI into road surface monitoring is novel, and the systematic evaluation across different proprietary and open-source models provides useful insights. The study is well motivated and relevant to both urban/transportation planning as well as AI research. But overall, the manusscript is not yet ready for publication. With clearer descriptions, stronger methodology, and better figure clarity, it could become a solid contribution to AI in transportation infrastructure. MAJOR REVISIONS REQUIRED.

Abstract:

23-25: Clear but the claim that GPT 4o provides the most favorable balance between accuracy and cost should be carefully stated

Introduction:

- HIghlights the importance of pavement condition assessment and the potential of generative AI, but it does not sufficiently discuss the study in relation to existing work.

- prior work in deep learning and computer vision for crack detection and PCI estimation mentioned but does not reference or contrast with these efforts in detail

- novelty of benchmarking multimodal generative models is not sharply presented

- The introduction would benefit from more explicitly stating how this study differs from prior research

Methods

- dataset size is small: needs justification

- 157–160: “ground truth” labels for maintenance interval based on temporal comparisons of Google Street View imagery, which may not reliably capture the actual timing or type of repairs. This introduces potential uncertainty in the benchmark labels

- 162-166: Manual annotation procedure is not described in enough detail. How many? Was inter-rater agreement measured?

- 205-208: definition of appears very subjective. any formal coding protocol or double-checking used?

- Statistical testing is ok, but effect sizes should also be reported to quantify practical differences, not jsut p values

Results:

- GPT-4o and OpenAI o1 are highlighted as top-performing, ut GEmma 3 is under-discussed even though it reached 100% response in one task, Being open-source, might be more important for reproducibility and cost.

- Hallucination analysis would benefit from systematic quantification across tasks and models

- the figures are difficult to read at this scale

Limitations:

- Actual pavement management requires more than just surface imagery?

- Fine tuning and prompt engineering are mentioned but not fully discussed in domain specific needs

- Comparison between Open source and proprietary could be expanded

6. PLOS authors have the option to publish the peer review history of their article (what does this mean?). If published, this will include your full peer review and any attached files.

Reviewer #1: No

Reviewer #2: No

Reviewer #3: No

---

## [Author Response · Author response to Decision Letter 1]

5 Nov 2025

we attached the responses in the document submission.

---

## [Decision Letter · Decision Letter 1]

12 Dec 2025

PONE-D-25-46997R1Multimodal Generative AI for Automated Pavement Condition Assessment: Benchmarking Model PerformancePLOS One

Dear Dr. Cui,

Thank you for submitting your manuscript to PLOS ONE. After careful consideration, we feel that it has merit but does not fully meet PLOS ONE’s publication criteria as it currently stands. Therefore, we invite you to submit a revised version of the manuscript that addresses the points raised during the review process.

We look forward to receiving your revised manuscript.

Kind regards,

Junghwan Kim

Academic Editor

PLOS One

Journal Requirements:

Additional Editor Comments:

I have received all the reviewers' reports and completed the evaluation of the revised manuscript. Reviewer #2 still has an outstanding item. Please address this. Thank you!

Reviewers' comments:

Reviewer's Responses to Questions

**Comments to the Author**

1. If the authors have adequately addressed your comments raised in a previous round of review and you feel that this manuscript is now acceptable for publication, you may indicate that here to bypass the “Comments to the Author” section, enter your conflict of interest statement in the “Confidential to Editor” section, and submit your "Accept" recommendation.

Reviewer #1: All comments have been addressed

Reviewer #2: All comments have been addressed

Reviewer #3: All comments have been addressed

2. Is the manuscript technically sound, and do the data support the conclusions?

Reviewer #1: Yes

Reviewer #2: Yes

Reviewer #3: Yes

3. Has the statistical analysis been performed appropriately and rigorously? 

Reviewer #1: Yes

Reviewer #2: Yes

Reviewer #3: Yes

4. Have the authors made all data underlying the findings in their manuscript fully available?

Reviewer #1: Yes

Reviewer #2: Yes

Reviewer #3: Yes

5. Is the manuscript presented in an intelligible fashion and written in standard English?

Reviewer #1: Yes

Reviewer #2: Yes

Reviewer #3: Yes

6. Review Comments to the Author

Reviewer #1: (No Response)

Reviewer #2: Many thanks to the authors for addressing my comments. I can see that the new version is well improved. All my concerns have been addressed. However, I recommend that the authors ensure that the claims added are adequately supported by appropriate references before the final version is published.

Reviewer #3: (No Response)

7. PLOS authors have the option to publish the peer review history of their article (what does this mean?). If published, this will include your full peer review and any attached files.

Reviewer #1: No

Reviewer #2: No

Reviewer #3: No

---

## [Author Response · Author response to Decision Letter 2]

17 Dec 2025

2025-12-17

Manuscript PONE-D-25-46997R1

Response to Editor and Reviewers

# regarding funding source

We have spelled out the CESU to Great Lakes–Northern Forest (GLNF) Cooperative Ecosystem Studies Unit (CESU) in the portal website as well as a statement in the cover letter: The funder had no role in the study design, data collection and analysis, decision to publish, or preparation of the manuscript. The content of this publication does not necessarily reflect the views or policies of the funder.

# regarding data availability statement

We have added a data availability statement in the submitted material.

# response to the Reviewer #2

Additional Editor Comments

I have received all the reviewers' reports and completed the evaluation of the revised manuscript. Reviewer #2 still has an outstanding item. Please address this. Thank you!

Authors’ response: Thank you very much for your time and for reviewing the revised manuscript. We have addressed the remaining comment from Reviewer #2. A detailed explanation of the revisions made in response to this comment is provided in our response to Reviewer #2 below.

Reviewers' Comments to the Authors: Reviewer

Reviewer #2

Many thanks to the authors for addressing my comments. I can see that the new version is well improved. All my concerns have been addressed. However, I recommend that the authors ensure that the claims added are adequately supported by appropriate references before the final version is published.

Authors’ response: We sincerely thank the reviewer for the positive assessment of the revised manuscript and for this helpful suggestion. In response to this helpful suggestion, we carefully reviewed the claims added in the previous revision and extended this review to the references supporting all claims throughout the manuscript to ensure appropriate scholarly support.

To further strengthen the academic rigor of the manuscript, we modified several references, particularly in cases where non–peer-reviewed sources were previously cited, by replacing them with appropriate peer-reviewed literature. When a claim was already supported by multiple peer-reviewed studies, the corresponding non–peer-reviewed reference was removed. When a claim relied primarily on a non–peer-reviewed source, we replaced it with an appropriate peer-reviewed citation. In cases where no suitable peer-reviewed reference could be identified to support a given claim, both the claim and the corresponding non–peer-reviewed reference were removed.

References removed

References 13, 27, 58, 59, 71, 73, 76, and 80 were removed.

References updated or replaced

• Reference 25 was replaced with:

Whang SE, Lee JG. Data collection and quality challenges for deep learning. Proceedings of the VLDB Endowment. 2020;13(12):3429–3432.

This supports the claim: “Acquiring such data is often challenging and time-consuming due to the need for extensive manual annotation.”

• Reference 28 was replaced with:

Bandi A, Adapa PV, Kuchi YE. The power of generative AI: A review of requirements, models, input–output formats, evaluation metrics, and challenges. Future Internet. 2023;15(8):260.

This supports the claim: “Unlike conventional machine learning models that require extensive training datasets and pre-defined rules, generative artificial intelligence (GAI) models can respond adaptively to a range of inputs and generate task-relevant outputs with limited contextual information”

• Reference 29 was replaced with:

Lu K, Zhao X, Li M, Wang Z, Zhou Y, Wang J. StreetSenser: a novel approach to sensing street view via a fine-tuned multimodal large language model. International Journal of Geographical Information Science. 2025.

This supports the claim: “Leveraging these capabilities, recent studies have begun to apply MLLMs in urban research, particularly using street-view imagery to analyze built environments, urban form, and neighborhood conditions.”

• Reference 30 was replaced with:

Blečić I, Saiu V, Trunfio GA. Enhancing urban walkability assessment with multimodal large language models. In: International Conference on Computational Science and Its Applications. Springer; 2024.

This further supports the same claim as reference 29.

• Reference 57 was replaced with:

Zhang S, Mu H, Liu T. Improving accuracy and generalizability via multimodal large language model collaboration. Proceedings of the IJCNN. 2024.

This supports the claim: “When baseline performance is insufficient, these models can be fine-tuned to improve task-specific accuracy.”

• References 68 and 69 were replaced with:

Wu J, Zhong M, Xing S, Lai Z, Liu Z, Chen Z, Wang W, Zhu X, Lu L, Lu T, Luo P. VisionLLM v2: An end-to-end generalist multimodal large language model for hundreds of vision–language tasks. Advances in Neural Information Processing Systems. 2024.

This supports the claim: “In contrast, MLLM can address multiple tasks within a unified framework by combining visual perception with linguistic reasoning.”

• Reference 96 was replaced with:

Kuang J, Shen Y, Xie J, Luo H, Xu Z, Li R, Li Y, Cheng X, Lin X, Han Y. Natural language understanding and inference with MLLMs in visual question answering: A survey. ACM Computing Surveys. 2025.

This supports the claim: “First, they allow users to interact with the model using natural language prompts, removing the need for specialized programming expertise.”

• Reference 98 was replaced with:

Zhou X, He J, Ke Y, Zhu G, Gutiérrez-Basulto V, Pan J. An empirical study on parameter-efficient fine-tuning for multimodal large language models. Findings of ACL. 2024.

This supports the claim: “One key area is the need for additional domain-specific training data to support fine-tuning and improve model performance.”

• Reference 99 has now been published as a peer-reviewed article, and the citation has been updated accordingly.

• Reference 100 was replaced with:

Chen B, Zhang Z, Langrené N, Zhu S. Unleashing the potential of prompt engineering for large language models. Patterns. 2025.

This supports the claim: “At the same time, prompt engineering plays an important role in improving model outputs, particularly for context-dependent tasks.”

In addition, to further enhance academic precision, we refined the wording of several claims while preserving their original meaning. For example, Reference 56 was replaced with Wu J, Zhong M, Xing S, Lai Z, Liu Z, Chen Z, Wang W, Zhu X, Lu L, Lu T, Luo P. Visionllm v2: An end-to-end generalist multimodal large language model for hundreds of vision-language tasks. Advances in Neural Information Processing Systems. 2024 Dec 16;37:69925-75 “Although MLLMs have not yet reached the point of full automation, they demonstrate versatility across tasks.”

Finally, after updating the references, we slightly reorganized the relevant paragraph on page 6 to improve clarity and logical flow. The revised paragraph now reads:

“Recent advancements in GAI have opened new avenues for automating assessment processes across various domains (54, 55). Although MLLMs have not yet reached the point of full automation, they demonstrate versatility across tasks (56). Traditional machine learning approaches for surface-level road condition assessment are typically designed and trained for a single, well-defined objective such as defect detection (60-62), defect classification (63-65), or surface segmentation (66, 67). In contrast, MLLM can address multiple tasks within a unified framework by combining visual perception with linguistic reasoning (68, 69). In addition, when baseline performance is insufficient, these models can be fine-tuned to improve task-specific accuracy (57).”

---

## [Editor Report · Decision Letter 2]

21 Dec 2025

Multimodal Generative AI for Automated Pavement Condition Assessment: Benchmarking Model Performance

PONE-D-25-46997R2

Dear Dr. Cui,

We’re pleased to inform you that your manuscript has been judged scientifically suitable for publication and will be formally accepted for publication once it meets all outstanding technical requirements.

Kind regards,

Junghwan Kim

Academic Editor

PLOS One
---

## [Editor Report · Acceptance letter]

PONE-D-25-46997R2

PLOS One

Dear Dr. Cui,

I'm pleased to inform you that your manuscript has been deemed suitable for publication in PLOS One. Congratulations! Your manuscript is now being handed over to our production team.

Kind regards,

on behalf of

Dr. Junghwan Kim

Academic Editor

PLOS One